# Challenges Caused by Increased Use of E-Powered Personal Mobility Vehicles in European Cities

**Jurgis Zagorskas \*** and **Marija Burinskienė**

Department of Roads, Vilnius Gediminas Technical University, Saulėtekio al. 11, 2510 Vilnius, Lithuania; marija.burinskiene@vgtu.lt
**\*** Correspondence: jurgis.zagorskas@vgtu.lt; Tel.: +370-688-14666

**Abstract:** Increased use of e-powered personal mobility vehicles is usually considered to be a positive change, while it is generally agreed that Personal Mobility Vehicles (PMVs) effectively and efficiently reduce the negative environmental impacts of transport and improve quality of life. There has been great technological progress made by all sectors in the field of personal mobility during the last decade. The use of PMVs for micro-mobility have been welcomed by the market, consumers, and governments and thus they are becoming increasingly popular in modern European society. New technology-driven PMVs provide opportunities to their users, but at the same time create problems with street space sharing, road safety, and traffic offenses. This study gives an overview of recent types of PMVs, offers some insights into upcoming changes and challenges, and raises a discussion on themes related to the increased use of e-powered personal transporters.

**Keywords:** urban sustainability; sustainable transportation; e-scooters; personal mobility vehicles (PMV); personal light electric vehicles (LEV)

## 1. Introduction

Urban micro-mobility has been taking fundamental changes in the last decade. In dense urban environments, individual automobiles are being regarded as an unsustainable mode of transportation, and there is a shift underway in policy together with pragmatic considerations of society in favor of eco-friendly, compact, and light vehicles. Reduced automobile use can help to achieve many strategic targets—cities can get rid of traffic jams, reduce greenhouse gas contributions, reduce noise levels, and improve their air quality. While the number of vehicles is rapidly growing, urban planners and transportation professionals are attempting to change people's travel mode selections towards less energy-intensive modes—walking, cycling, and similar [1]. Electrical power-assisted personal mobility vehicles (e-PMVs) represent a relevant alternative—they are a convenient and environmentally friendly mode of transportation for short trips in cities that can otherwise be clogged up with traffic [2].

E-PMVs are cost-effective only on short-distance trips between 0.8 and 3.2 km, which is where e-PMVs would be a particularly strong alternative to private automobiles [3]. Estimations of recent urban travel mode selection changes in the USA show that e-scooters can also replace up to 1% of taxi trips in central city areas [4]. By filling this gap in mobility, e-PMVs have the potential to decrease car use, but due to their higher relative cost on longer trips, e-PMVs would likely not result in a significant diversion away from public transit on longer-distance trips, particularly services operating to and from jobs. The use of e-PMVs on these longer journeys would likely be short connections to nearby transit stops [2]. However, the observed use of e-scooters in connection with transit is small due to the relatively high additional costs of doing so [5].

While e-scooters offer an alternative to automobiles (34% of riders surveyed said they would have made the trip by car had the devices not been available), other studies show that with an even

higher percentage (45–49%) of riders would have walked or biked instead [2,4]. Recent research shows that e-PMVs are partially replacing public transport, walking, and cycling [6]. Research conducted in Spain, Europe shows that e-PMV usage can be still considered minor within the whole sample of use of different means of transport in urban trips. Individuals using shared mobility services more than twice per week represent 3.5% of Spain's total population. There are still many more people often walking on foot (58%), traveling by public transport (53%), and using private cars (34%) [7].

According to market analytics [8], the global PMV market which includes scooters, walking aids, wheelchairs, and other similar vehicles is expected to grow at least until the year 2024 at a compound annual growth rate of 7.0%. This number shows unprecedented potential. North America is now the leading market for PMVs. Europe is now in second place and accounts for around 35% of PMV market share [9].

Technological advancements undoubtedly revolutionize urban mobility, commuting, and our way of life. However, European society has reached a stage in which the fascination with technological innovations often results in their indiscriminate consumption [10]. Shared e-scooters, as an example, demonstrate that the introduction of technology provides some benefits for their users, but at the same time challenges other parts of society. It is not enough to introduce the technology for mass consumption—work must be conducted to make the most of its potential and prevent unfavorable outcomes, developing the adequate infrastructure for its proper usage.

The aim of this paper is to outline and systematically describe the emerging issues in e-PMV implementation. This paper raises the discussion on what measures are needed for the reorganization of public and street space and overviews current policies on PMVs.

## 2. E-Powered Personal Mobility Vehicles

The e-PMV is a small personal vehicle that runs on electricity. It is powered by rechargeable lithium-ion batteries, and can travel at speeds of 20 to 60 km/h. PMVs nowadays include e-bicycles, e-scooters, mono-wheels, self-balancing devices, and other devices like e-skateboards (Figure 1). E-PMVs are frequently used for short-distance trips and are becoming common in urban spaces. Field observation data gathered by the authors of this paper in several European cities for 2019 show that e-PMVs are equally popular or more popular than the bicycle. In central Paris and Barcelona, they take up 45–60% of all non-motorized and e-PMV traffic volume. A similar situation occurs in the cities less frequently visited by tourists that have public e-scooter sharing services (Warsaw in Poland, Vilnius, Kaunas in Lithuania). This situation will probably not change in the near future. With developed technologies—the higher efficiency of electric motors, the bigger capacity of batteries, integrated computer processor units, and powerful lighting—e-PMVs have become strong competitors to traditional bicycles. Future PMVs should be energy efficient, compact, light, easy to carry by hand, safe to drive, and easy to handle. Availability for sharing is also important, which is enabled by GPS tracking and the use of smart-phones [11].

Vehicles with electric assistance or fully e-powered vehicles can perform at the level of a professional sportsman. A regular cyclist can generate up to 350 W of power and the average generated power at a cruising speed is 100–160 W. A person with high physical preparation (a sportsman) can generate two times more power—an instant 700–1000 W of power and an average of 250–350 W of power. Light electrical engines mounted on modern e-bicycles, e-scooters, and similar devices usually add or operate with the power of 350–1000 W. At the same time the additional weight of the motor and batteries (5–15 kg) adds up, therefore making the performance less dramatic. E-power assistance increases the average travel speed two or more times from a regular 12–17 km/h bicycle speed to a 25–50 km/h e-motor speed [12]. A 50 km/h speed is achieved only by the most powerful devices with 750 W or more. In many cases the maximum speed is limited electronically for safety reasons.

Vehicle characteristics and the operating speed range shown in Figure 1 reveal important alterations of mixed vehicle types. In the past, when the streets were dominated by pedestrians, bicycles, and car traffic, three very apparent speed zones were present (4–7 km/h for pedestrians, 15–25 km/h for bicycle

users, and 30–50 km/h for car users). With the appearance of e-PMVs, the operating speed of these types of vehicles has gradually risen and is becoming similar to car traffic speeds, as seen in the last column of Figure 1. This creates implications for street space sharing and increases the chances of accidents occurring.

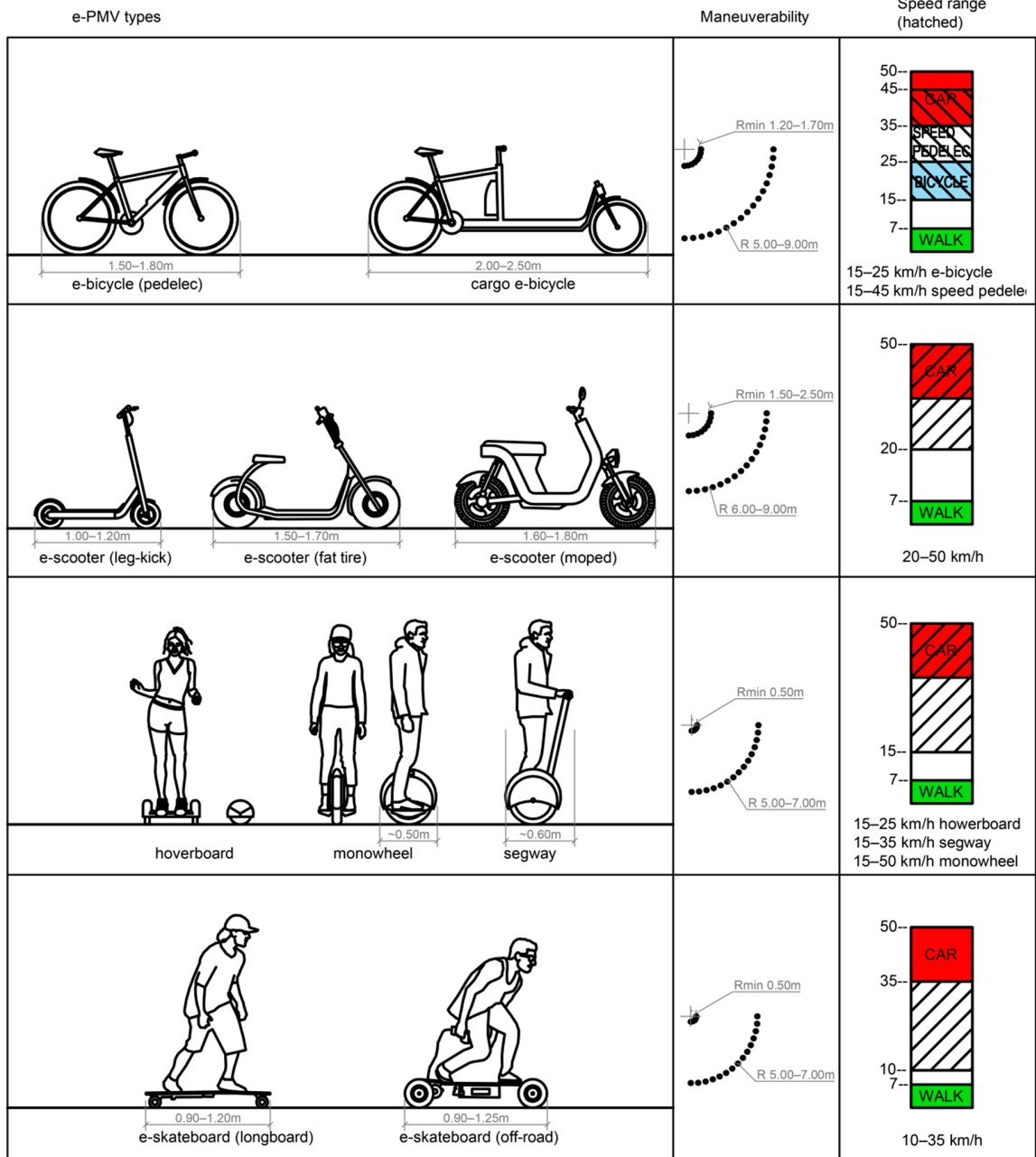

**Figure 1.** Types of most popular e-PMVs, their maneuverability, and operating speeds.

E-powered PMVs, also called mopeds or scooters and speed pedelecs in Europe, are characterized by their maximum continuous rated power of no more than 4 kW and a maximum speed of 45 km/h, are defined in the EU-Regulation No 168/2013 as vehicle class L1e-B. Both motorcycles and PMVs fall in this category [13]. PMVs, in general, accounted for 11.5% of the European passenger-vehicle mix in 2014 [14]. By now this number is likely significantly higher.

E-scooters are now the most popular e-wheelers in European cities. Field observation data show that they have already overcome traditional bicycles by popularity in some cities. A leg-kick e-scooter

has several advantages, including its small size and the fact that many devices are foldable, making them easy to transport in a car or public transport. The standing position of these scooters makes it easy and quick to jump on and off them, and provides the option to perform many daily activities just by standing on the vehicle. On the other hand, riding an e-scooter at a higher speed can be dangerous—a collision sustained in an erect standing position often results in serious injuries to the hands and head. A "fat-tire" e-scooter has wide tires and can travel smoothly on uneven surfaces or stand on the road without support. It is more comfortable and safe to ride than a leg-kick e-scooter, and is very smooth and quiet during a ride. Moped type e-scooters can be considered in the same category as combustion engine mopeds and often use the same space with the cars, since their motors are usually more powerful and their batteries are bigger. It can be considered to be safer than a leg-kick type e-scooter due to requiring a sitting position and the compulsory use of a helmet.

With the invention of a self-balancing mechanism, several types of devices appeared on the streets starting from the Segway brought to market in 2001. The same mechanism was later introduced in many types of e-wheelers. The most recognized e-wheelers at present are the "hoverboard" and "monowheel" devices. "Hoverboard" vehicles are popular with children and teenagers and mostly are used for fun in playgrounds, parks, etc. The "monowheel" is recognized for its compact size (it is possible to carry in a regular bag in public transport, inside buildings, etc.), elegant and smooth riding experience, high speed (750 W–1 kW models can reach up to 50 km/h), and good maneuverability. The disadvantages are the higher price and the fact that it requires some time (usually a few days) to learn how to handle this device.

Electric motors are being integrated into many types of leisure or sporting devices such as skateboards. Powered models come with modifications for smoother riding on uneven surfaces. These modifications have bigger inflatable wheels. They are more powerful and usually have wired or wireless remote control for the user to utilize.

## 3. The Rapidly Growing Trend in Specific European Countries

Traditionally in many European cities (in France, Italy, Spain, Netherlands, Denmark, Sweden, etc.) bicycles were used for many short local trips of up to 7–10 km (15–45%). The bicycle still has unbeatable value as a mode of travel—it is cost-effective, simple and cheap, and it provides physical activity and therefore is beneficial to a user's health. However, the performance of the bicycle depends on the physical ability of the rider and the rider's willingness to expend all the energy needed to reach their destination [15]. The provision of power assistance to the rider expands the role of the bicycle and other PMV in urban transport. Added electric power assistance helps to cover bigger distances, ride at higher speeds, and cope with natural barriers like big inclines, windy, and hilly areas.

In recent years as many electric PMVs became available, the most popular in Europe became e-scooters, because of their ease of use and handling. It is believed that e-PTVs can help European cities to ease their problems with traffic, emissions, and parking [14]. In the years 2018–2019, the experiment with shared e-scooters was taking place in many European cities. It still continues and gets controversial feedback in the media—these experiments generate many complaints from the public and raise questions about pedestrian safety and the impact of the devices on public spaces—alleys, avenues, squares, and parks [16,17].

The development of the PMV industry is cultivating many changes in mobility behavior [18]. In recent years (starting from 2015) e-PMV usage increased greatly in Europe. Analysis completed by the Innovation Centre for Mobility and Societal Change (InnoZ) GmbH shows that between 2016 and 2017, the number of scooters quadrupled (with 350,000 registered users in 2017) and from between 2017 and 2018, the number of scooters almost tripled. The countries where the fastest growth of private e-scooter use has been observed in the year 2019, are: Spain (498% growth per year), France (132%), Germany, Italy (286%), Poland, Austria, Netherlands, Belgium, and Switzerland [9]. A total of 55% of all e-scooters are in Spain and France alone. E-scooter sharing was available in most the European countries by the year 2019. More than every second scooter remains distributed within

Spain and France, which are the two dominant markets. Germany is currently in third place. The main European cities flooded by shared e-scooters today are Madrid, Paris, Barcelona, Berlin, Milan, Rome, and Nice [8,9].

Many car-sharing service providers in Europe also started to offer e-scooters recently. For instance, Swiss Mobility is active in both scooter-sharing and car-sharing. The Austrian automobile and mobility association ÖAMTC started to offer electric scooters in 2018. This trend is continuing in 2019 [9]. The e-scooter sharing sector still lacks scientific descriptions of user characteristics, as it is only known that the majority of users are young. Scooters are often used for commuting or leisure time activities [19]. E-bicycles and e-scooters are used for smart logistics in some cases in Europe [20], but this trend is only starting to develop.

## 4. The Impacts of PMVs on Existing Transport Systems

To be able to predict and respond to the changes in micro-mobility, it is important to understand and analyze the situations when users prefer PMVs instead of other transport means. Field observation shows that e-PMVs are popular amongst children, teenagers, and adults and are slightly less so for elderly people. There can be several reasons for this popularity: the natural tendency for people to search for more convenient types and modes of travel in congested cities; a number of new inventions in electric motor efficiency, prolonged battery life, and energy storage capacity; increased micro-mobility without physical effort; a significant increase in fossil fuel prices; and other rationales.

Data obtained from e-scooter sharing companies from Vilnius city, Lithuania shows that the distance for e-PMVs rides tends to be 4–5 km with an estimated travel time of between 15 and 20 min. These values can differ from city to city [9]. Walking and cycling were traditionally the most common modes for micro-mobility, i.e., traveling short distances for daily needs. These modes are now being partially replaced by e-PMVs, with the biggest portion taken by e-scooters.

According to EU Commission recommendations, the distance to the destinations with some of the most important urban functions are:

| Objects: | Distance, m | Dominating transportation mode |
|---|---|---|
| *Neighborhood scale* | | |
| Public transport stops; | 300 | Walking, PMV |
| Children playground; kindergarten; | | |
| Preliminary school; | 400 | Walking, PMV, bicycle, car, public transport |
| Local shop, food store; | | |
| *District scale* | | |
| Sport club, park; | 600 | Walking, PMV, bicycle |
| Health care centers; | 800 | Car, public transport, PMV, bicycle |
| Secondary school; | 1000 | Walking, Car, public transport, PMV, bicycle |
| Leisure centers; | 1500 | Car, public transport, PMV, bicycle |
| Supermarkets; Hospitals; | 5000 | Car, public transport, PMV, bicycle |

E-PMVs can serve to reach all these destinations.

## 5. Adaptation of Existing Infrastructure for e-PMVs

In road traffic regulations, e-PMVs are now considered in the same category as bicycles. A bicycle network is the part of a town transportation system and in great part, it shares the same infrastructure with cars and pedestrian sidewalks, while having the same travel origins and destinations. PMVs share the same infrastructure with bicycles and cars—pathways, parking spaces, etc. If some of the car and bicycle users in the future will change to PMVs, the remaining infrastructure will have some mismatches as Table 1 shows. Space which is now used by cars can be reorganized to serve public transport and e-PMVs together with bicycles. On the other hand, the pavements and bicycle paths can become more crowded, but it will not create big problems, because of the tiny space used by PMVs compared to an automobile.

**Table 1.** Typical travel distances, speed, and other parameters of different coexisting modes of transportation in European cities.

| Travel Mode | Typical Trip Distance, km | Average Travel Speed, km/h | Used Space, $m^2$ | Maneuvering Radius in Minimal Speed, m | Maneuvering Radius in Cruising Speed, m |
|---|---|---|---|---|---|
| Walking | <1.5 | 4–6 | 0.5–1.0 | 0.0 | 0.5 |
| Cycling | 0.5–8 | 12–15 | 1.2–1.6 | 3.2–4.0 | 8.0–12.0 |
| E-bicycle | 0.5–15 | 15–35 | 1.2–1.7 | 3.2–4.0 | 12.0–18.0 |
| E-scooter (leg-kick type) | 0.5–5 | 15–25 | 0.8–1.2 | 1.5–2.5 | 1.5–2.5 |
| E-scooter (moped) | 1–20 | 20–40 | 1.2–2.0 | 3.5–5.0 | 16.0–20.0 |
| motorcycle | 1–20 | 25–50 | 1.5–2.2 | 3.5–5.0 | 16.0–20.0 |
| Public transport | 1–20 | 30–35 | 0.5–1.0 | 6.0–9.0 | 50.0–90.0 |
| Car | 2–35 | 35–50 | 5.0–12.0 | 3.5–6.0 | 40.0–50.0 |

Recently with growing concerns about public health and safety issues, the revival of separating bicycle paths from motorized traffic has been observed [21]. The latest research topics show that separating bicycle lanes from main motorized traffic volumes for health reasons is an important and highly recognized strategy [1,22,23]. The importance of relocating urban bike lanes to the calmer streets or nature predominated places has been stressed by many researchers [24]. The bicycle networks are planned to be separated and moved away from heavy traffic to safer and healthier environments like unused riversides, parks, and natural territories [25–28]. When implementing these measures, it must be taken into consideration that this infrastructure will be shared with e-PMV users, therefore new lanes must be designed according to requirements for higher speeds and, if possible, with divisions for walking, low speed, and higher speed lanes.

At the same time in dense urban environments there is another popular measure of setting low-speed zones (30 km/h) when no division of space is needed, which makes driving a car inconvenient. This measure is often used in town centers, historic districts, or places with dense and narrow streets.

Researchers nowadays are focusing on studying cyclist and PMV user behavior and analyzing what makes these travel modes comfortable and pleasant with the aim to set the rules and provide a friendlier environment for this kind of travel. During the past decades, a number of methods have been endorsed for the selection of suitable bicycle routes including meta-heuristics, Q-learning algorithm, and others [29–32]. Many findings of cyclist preferences and behavior were collected from smartphones [33–36] and BIG-data sources [37–39]. These methods were focused mostly on bicyclist travel route preference in an existing network with the aim to predict the travel demand and highlight safety problems. Some of these findings could also be used to enhance solutions for developing the existing bicycle and PMV pathway infrastructure. The findings show that each individual cyclist or PMV user may have different priorities between travel time and suitability when they choose a route, mostly because they travel on two to three routes within the same origin-destination pair and the perceived safety of that environment is the most important factor [32,40,41].

Due to interest in eco-friendly travel modes and EU funding, there has been a significant increase in the quality of pavements, crossings, safety measures, lighting, and other infrastructure in the last decade. The general urban population is starting to recognize the benefits of these investments and recently developed pedestrian and bicycle infrastructure for e-PMVs as well. But in general, many e-wheelers have small wheels to make them foldable and compact. A small wheel size requires very smooth surfaces and precisely leveled edges that are not yet present in many parts of European cities, especially in historic street pavements. It creates discomfort for the rider and can increase the danger of falling down and sustaining an injury. Some of e-PMVs are not so sensitive to pavement quality and have characteristics similar those of the to e-bicycle wheel. Between these are e-mopeds, fat-tire e-scooters, monowheels, segways. The relationship between the smoothness of surface, beveled edges of crossings, etc. has to be studied in more detail through separate research.

## 6. PMVs and Road Safety

Before permitting PMVs in shared environments, the impacts of the PMVs on the other users of the shared space should be properly evaluated, particularly from a safety perspective. Studies show that injuries occurring from these types of vehicles are significant and seem to be increasing [42]. E-scooters of the leg-kick type cause more injuries than bicycles and other types of common e-wheelers (e-bicycles, e-mopeds, and self-balancing devices) because they involve traveling at high speed in the standing position and because of their availability through public sharing services for users without any riding experience.

The popular research trend today involves exploring the safety of bicycle (and at the same time PMV) routes [43–45]. Many researchers have put efforts into establishing methods to explicitly address bicycle safety by reflecting urban conditions, and have found that many factors influence safety, including the traffic volume, lane width, population density, highway classification, presence of vertical grades, one-way streets, and truck routes [21]. These urban conditions were taken into account to predict the severity of an injury that would result from a motor vehicle crash that can occur at a specific location [46–49]. The same methods and findings can be applicable to e-PMV safety. E-PMVs also have impacts on other shared transportation space users. These impacts are not limited to dangers from riding, as the risk can also be increased by taking the space provided for other vehicle types or through improper parking. In Singapore, where the use of PMVs has a longer history and tight rules for usage in public spaces [50], the conducted studies [51] show that motorized PMVs tripled the risk of severe injury and doubled the risk of requiring hospitalization, compared to non-motorized personal mobility devices, due to higher traveling speeds [52]. A similar study from Korea shows that the number of injuries related to PMVs in 2017 increased three times compared to the year 2016 (51 vs. 14). Injuries to the head and neck region (67.7%) were the most common types, followed by upper extremity injuries (46.2%) [53].

From these studies, it can be concluded that further investigation into the risks of their use and training or practice, particularly for new or infrequent users (such as tourists), is needed. Protective equipment such as helmets should also be considered. Several previous studies investigated the operational and safety aspects of PMVs in mixed traffic conditions. One of the most relevant studies focuses on pedestrians' danger perception toward PMVs when interacting with them in shared spaces by estimating a subjective danger index (SDI) [54]. In this study on the safety of shared space, the concept of safety was divided into objective or physical safety and subjective or psychological safety. Psychological safety has great importance and is one of the main reasons why the general public has been shocked by a sudden invasion of e-PMVs. The power of the e-motor exceeds regular human power by up to 2–3 times and can cause serious danger if this vehicle is on the same path or lane with pedestrians or even regular cyclists. Changes in vehicle power change speed, momentum, maneuverability, driving trajectory parameters, and future bicycle lanes or road infrastructure must be adapted to these changes.

When the pathways are designed for 30 km/h speeds, the pedestrians must be separated from these pathways with a minimum separation distance of 1.50 m. If in the future the pathways for e-PMVs would be designed for 50 km/h speeds, then speed cyclists must be separated from these paths.

Amongst the professionals, there are proposals to limit the speed of shared e-scooters to a safe maximum. As a softer measure, speed limitations can be easily placed on many electrically driven devices with computer processor units. E-scooters can have imposed speed limits or even geo-fencing in dockless e-scooter applications to provide safer speed zones in certain parts of the cities (geo-fencing reduces the speed of e-scooter automatically when it enters a certain area).

## 7. Current European Policies on Improper Parking and Riding Behaviors

European transportation planning policy in most countries favors light transportation modes such as bicycles and vehicles under EU category of electric light vehicles [55]. Bicycling, walking, and similar modes contribute zero greenhouse gas emissions, and therefore promoting these modes

helps to keep promises on climate change mitigation. As an examples of this, the EU HORIZON 2020 project ELVITEN (the full name of the project is "Electrified L-category Vehicles Integrated into Transport and Electricity Networks") was launched in 2017 (mostly in Italy) to demonstrate how electric light vehicles (EL-Vs) can be used in urban areas and integrated into the existing transport networks of six European cities. Another example is the already finished Pro-E-Bike project (www.pro-e-bike.org), funded within the Intelligent Energy Europe program. The project investigated the potentialities of e-bicycles and e-scooters for goods delivered and services provided in urban areas [20].

European cities are relying on rules previously made for bicycles, rather than ones that take into account the unique aspects of e-PMVs. In most European countries, the user does not need a driver's license or a helmet to use a scooter-share vehicle. In many southern countries (Italy, France, Spain, etc.) there's also a long-standing habit of using mopeds, which is making the idea of low-powered motor vehicles habitual for locals.

Europe has been undergoing the experiment of a dockless e-scooter introduction to public spaces since 2016. E-scooters are a new type of vehicle and gained popularity so quickly that they are largely unregulated. The regulations currently being implemented are focused more on the scooter companies than on users [56]. What is happening now is that the private companies are profiting off of taxpayers' investment in sidewalks and making them less useful for residents. In response to that, many e-scooter sharing initiatives were restricted: France banned electric scooters from pavements in September 2019 to stop them from invading pedestrian areas [57]; in December 2018, the Madrid city government ordered the main operator companies to remove their scooters from the streets, saying they had failed to comply with rules that determine which areas the scooters are allowed to operate in [58], while Spanish tourist hotspot Barcelona has banned scooter rental services completely [59]; Berlin's city hall has also drawn up tough new rules for e-scooters [60]. In Germany, e-scooters must be limited to a speed of up to 20 km/h. For riders, helmets are not mandatory but are recommended. E-scooters in Germany may be ridden by individuals who are 14 or older. They need to use bike lanes. If there are none, they need to travel on the road [61].

To deal with improper parking and riding behavior in Paris, anyone riding any type of e-PMV on the pavement has been fined 135 euros since September 2019 [57]. E-PMVs have to use the street or dedicated cycling paths. The speed or e-PMVs is limited to 20 km/h in the city and 8 km/h in pedestrian areas. Paris officials plan to regulate the use of e-scooters with fines for riding on the sidewalks, designated parking spots, and an annual fee for e-scooter operator companies. During every upcoming year, the Paris municipality will create 2500 new parking spaces for e-scooters [62]. Several European cities already provide dedicated areas for scooter parking.

In 2019, Sweden banned the use of any motorized scooters capable of reaching speeds beyond 20 km/h from its cities' bicycle lanes [63,64]. A speed limit for the vehicles has been introduced in Belgium, where they can be ridden by anyone aged 18 or over under the same laws as bicycles, although the country recently raised the speed limit from 18 km/h to 25 km/h [65]. Some lessons can be learned from experiences in the USA over the last years. The manufacturing and first sale of an e-PMV is regulated by the federal law where it is defined as "A two- or three-wheeled vehicle with fully operable pedals and an electric motor of less than 750 watts (1 horse power), whose maximum speed on a paved level surface, when powered solely by such a motor while ridden by an operator who weighs 170 pounds, is less than 20 mph." However, e-PMV operation on streets and bikeways lies within a state's control rather than that of the federal government. Thus, states have their own laws that categorize e-PMVs, require licenses and registration, or do not enable them to be used on facilities such as bike lanes. State legislation in the US from the year 2015 has focused mainly on revising older state laws, creating three-tier classification systems for e-PMVs depending on their speed capabilities and refining more recent e-PMV laws. A three-tiered e-PMV classification system differentiates between models with varying speed capabilities. Then, based to the class of vehicle, restrictions are applied in many different ways across the USA [66]. Municipalities are starting to implement light regulations such as a minimum rider age, banning the vehicles from important recreational areas, and removing

them from places where they block ramps for wheelchair users [67]. There can be different measures to deal with arising problems. E-wheelers can be promoted as a micro-mobility revolution that serves the general public, but can also be restricted wherever they cause safety issues.

## 8. Conclusions and Discussion

There has been great technological progress in the e-PMV sector. Traditionally in European cities the bicycle facilitated many (15–45%) short local trips, but now many of these trips are made by using e-PMVs. Added electric power assistance help riders to travel longer distances, ride at higher speeds, and cope with natural barriers like large inclines, and windy or hilly areas.

The use of e-PMVs can lead to a cleaner, more sustainable future where e-powered fast and compact vehicles would be the primary mode of transportation but it can also have negative side effects, including: decreased road safety for PMV users themselves and other traffic participants; misuse of easily affordable dockless devices; and environmental problems, such as the mass e-scooter graveyards and discarded bike-share bicycles recently seen in China [68].

### 8.1. Potential Development of Infrastructure for e-PTVs

The development of the PMV industry is causing many changes in mobility behavior. The power of e-motors exceeds regular human power and can cause serious danger if this vehicle shares the same street space with pedestrians or cyclists. Added vehicle power change speed, momentum, maneuverability, and driving trajectory parameters are other factors that must be considered. Future bicycle lanes and infrastructure must reflect these changes. To develop suitable infrastructure, the first step should be to localize the most problematic street sections and crossings. For this purpose, GIS kernel density, geo-statistical and proximity analysis, and land-use regression models can be used. Network constrained KDE has been applied in a variety of disciplines, including traffic incidents and cycling infrastructure planning [43,69–71].

The speed of e-PMVs is greater than that of a bicycle and can reach 50 km/h. This fact creates implications for street space sharing and increases the risk of accidents. The straightforward solution is to separate transportation modes by providing isolated spaces for pedestrians, bicyclists, and e-PMVs with speeds of up to 30 km/h. But there can be many situations where this measure cannot be implemented. In such cases, e-PMVs can share the same space with public transport if public transport lanes are separated. Another option is to allow e-PMVs to use all the street space provided for cars in restricted 30 km/h speed zones.

Demonstrated methods to localize problematic areas and predict PMV volumes and the use of GIS tools can be used to monitor urban mobility changes, and help to develop of adequate infrastructure. GIS tools can be used by town and transportation planners and provide supporting arguments for street space reorganization and reconstruction, with positive implications for safety measures.

### 8.2. Suggestions for Policy-makers and Future Research Directions

Various policy measures can be applied to restrict and regulate the use of e-PMVs. European cities are now relying on rules previously created for bicycles, without taking into account the unique aspects of e-PMVs. In most European countries, the user needs neither a driver's license nor a helmet to use an e-PMV. This situation can be corrected at least partially by defining e-PMV classifications and placing restrictions on more powerful devices. Another important issue is that the accidents and traumas in which e-PMVs are involved in should be registered under separate categories to provide information for research on safety. If these initial issues will be fixed, there will be a solid basis and arguments for the improvement of infrastructure and for introducing other necessary measures.

There can be additional regulations like restrictions based on the amount of electric assistance provided by the vehicle. A vehicle with 250–300 Watts cannot provide a significant increase of speed and such a device can considered to have a similar speed to a bicycle. On the other hand, vehicles with 500–750 Watts can reach 50 km/h speeds and must be considered to be similar to a moped or even to

an automobile. In hilly areas, the assistance power of these vehicles can be larger to help climb the bigger inclines, but in in other cases, electronic speed limiters must be introduced. Devices with higher speeds could be banned from sidewalks and bicycle paths and could require a driving license. Devices with lower power could be used in the same manner as a regular bicycle.

There have been many attempts to regulate PMV sharing operator companies. This initiative was initiated by municipalities and such regulations are usually fines for improperly parked e-PMVs. These measures are becoming common practice, but in the future operator companies also should have obligations to place geo-fencing on dockless e-PMVs in town centers, parks, and squares. Until public awareness grows, the private users of PMVs could also be fined for improper riding and parking behaviors.

**Author Contributions:** Conceptualization, M.B. and J.Z.; Writing-Original Draft Preparation, J.Z.; Writing-Review and Editing, J.Z. and M.B.; Illustrations, J.Z.; Supervision, M.B.; Project Administration, M.B. All authors have read and agreed to the published version of the manuscript.

**Funding:** This research received no external funding.

**Conflicts of Interest:** The authors declare no conflict of interest.

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
