# Peer review of "Challenges Caused by Increased Use of E-Powered Personal Mobility Vehicles in European Cities"

_sustainability, doi:10.3390/su12010273_

Round 1

Reviewer 1 Report

This paper offers an excellent synthesis of the literature and offers a great deal of useful perspective on the tradeoffs associated with electric powered personal mobility vehicles (PMVs), particularly e-scooters.  The paper offers one of the most comprehensive overviews of the strength and weaknesses of PMVs I have read in an academic piece.  The article does a particularly good job summarizing what can be learned from previous literature that is informed by data from PMVs in major cities.  It is also well organized. 

I have one major recommendations and two minor ones to make this article suitable for publication. 

Major recommendations

1.  The section "Demonstration Study" provides analysis using kernel density analysis to show concentrations of PMVs, predictions about traffic, and conflict areas.  However, the reader is told very little about how this analysis was calibrated, what data was used, and how the results may depend on the assumptions made.  It is presented as in a very "matter of fact" way without acknowledgement of the many variables that may affect the results.  This appears to be the most significant original analysis included in the study, yet there is little information provided about all the uncertainty and complexity associated with making these estimates.  

I'm no expert on kernel density analysis.  However, it wasn't clear to me if analysis provided (including the maps) was for a particular city or whether the analysis was entirely hypothetical.  I wondered how sensitive the predictions in Figure 3 were to assumptions used in the model.  I wondered if these results had been previously published since so little was provided about the inputs into the model.  

This section ends without any real follow-up discussion about how this type of analysis could be used in different contexts.  Perhaps five to seven more paragraphs of discussion should be added to this section, perhaps by scaling back other parts of the paper.  Sensitively analysis should be provided, if that is appropriate using kernel density analysis, to showcase this analysis and its potential use in different settings. 

Minor items.

1.  The paper should have a clearer statement of purpose.  After I read the "this paper" paragraph on line 58, I still wasn't sure if the paper included original analysis or whether it was a completion of previous research.

2.  The sources or the methods used to compile Table 2 and 3 should be provided

3.  Some editing is needed.  There are places where the word "the" needs to be added to the beginning of sentences.  A partial list:

Line 58.  Add an "s" to the word environment
Line 117.  Add the word "a" before the word few
Line 250.   The word "study" should be "studies"
Line 252.  Add the word "The" before study
Line 281.  Add the word "The" before tool 

These are just examples.  A good editor to easily fix these and other problems.  It the authors can address the above issues, this article would be much improved and worthy of consideration for publication. 

Joe

Author Response

Response to reviewer

Thank you for your time and all advices given to improve our research paper.

Your insights were very valuable for us to understand weaknesses and strengths of our presentation of material.

In response to Your comments we conclude that the section "Demonstration Study" was totally removed from the paper, according to recommendations of other reviewers. This section will be detailed and presented for publishing as separate topic, within the category of “case study”.

Regarding minor items, we tried to make the statement of purpose clearer, Tables 2 and 3 were removed from paper according to other reviewer opinions, the editing will be done after the text go throw the second round. We apologize for mistakes, yet we were not able to find editor within a given period of 10 days for revision.

Thank you again for reviewing our work!

Reviewer 2 Report

The rapid growth in the use of electric scooters in many cities round the world makes this paper of potential value to researchers and policy makers around the world. This paper provides a useful characterisation of the range of electric personal mobility vehicles and Table 1 is particularly effective. The paper flags a range of very relevant issues in relation to these rapidly emerging new form of mobility but the discussion needs to be better grounded on evidence. There are many statements made in the paper which are not supported by evidence and therefore appear to be the opinions of the authors. A more valuable discussion paper would provide a deep examination of key issues, highlighting alternative interpretations and perhaps areas of agreement or disagreement based on evidence. 

The demonstration study outlined in Section 7 does not materially add to the paper. The methods are inadequately explained and there is no evidence of convincing validation of the model to provide reassurance of its usefulness for planning or policy analysis. It is suggested that section 7 be deleted from the paper and separate effort made to enhance the formulation and validation of the model so that it could be the focus of a separate paper.

Detailed comments are included in the attached annotated version of the paper. Addressing those comments would help to substantially improve the quality of the paper and ensure that it provides an evidence based contribution to this important topic. 

Author Response

Response to reviewer

Thank you for your time and all advices given to improve our research paper.

Your insights were very valuable for us to understand weaknesses and strengths of our presentation of material.

In response to Your comments we conclude that the section "Demonstration Study" was totally removed from the paper, Tables 2 and 3 were also removed.

“Demonstration study” part will be detailed and presented for publishing as separate topic, as case study.

The references were added where possible, the vague statements and speculations were removed or paraphrased.

References will be checked out and corrected after the second review round .

Thank you again for reviewing our work and valuable remarks!

Reviewer 3 Report

Anticipate the definition of PMV in the introduction  of the paper.

Add recerences in particular for fig. 1, line 76, 97, 142, 162, 182.

Aspects related to technical parameters (especially in ch. 6) are strongly related to specific national rules.

More work on existing regulations in the different countries  should be done before proposing new parameters.

I would suggest to add in ch. 6 an overview of existing regulations and do not propose specific parameters.

Author Response

Response to reviewer

Thank you for your time and all advices given to improve our research paper.

Your insights were very valuable for us to understand weaknesses and strengths of our presentation of material.

In response to Your comments :

>Anticipate the definition of PMV in the introduction  of the paper.

Definition added

>Add references in particular for fig. 1, line 76, 97, 142, 162, 182.

References were added

>Aspects related to technical parameters (especially in ch. 6) are strongly related to specific national rules. More work on existing regulations in the different countries  should be done before proposing new parameters.

>I would suggest to add in ch. 6 an overview of existing regulations and do not propose specific parameters.

The tables 2 and 3 with technical parameters were removed, the text corrected accordingly. Table 1 was left as it was, the numbers provided here can give  some insights for the readers.

Also the section "Demonstration Study", according to recommendations of other reviewers,  was totally removed from the paper, this part will be detailed and presented for publishing as separate topic, as a case study.

References will be checked and corrected after the second review round.

Thank you again for reviewing our work and valuable remarks!

Round 2

Reviewer 1 Report

I could not find documentation that other reviewers (at least one) recommended that the "Demonstration Study" section be entirely deleted.  I felt this was an important part of the paper that needed enhancement.   Based on margins comments from one other reviewer I did see, at least one other review felt the same.  If someone can provide documentation that another reviewer felt strongly this should be deleted, and that the paper was publishable without it, please share details. 

Author Response

Dear reviewer,

regarding removing the "Demonstartion study"  from our paper we were responding to reviewer 2 comments.

"reviewer 2" in first round have suggested that demonstration study
should be removed from our manuscript:
> "The demonstration study outlined in Section 7 does not materially add
> to the paper. ......
>
> ......... It is suggested that section 7 be deleted from the paper and
> separate effort made to enhance the formulation and validation of the
> model so that it could be the focus of a separate paper."
>

Else the article was given to extensive editing and it is now improved both by adding the missing references and correcting the style.

Your earlier comments helped us very much to understand what we need to finish the description of our case study project.

Thank you for your time and efforts, and great help!

Reviewer 2 Report

The authors have clearly taken on board the earlier feedback. There are some minor typos to correct in the revised version and some reference details to be added. Some questions have been inserted in the annotated copy of the paper for the author's reflection and they may result in minor edits. 

Author Response

Dear reviewer,

all your comments were considered and changes were made accordingly, as you can find in the last version of the uploaded document.

Thank you again for Your kind efforts and dedicated time to help us to improve this article!

Round 3

Reviewer 1 Report

Thank you for the clarification from the other reviewer about eliminating the analytical section. 

Author Response

Dear reviewer,

minor spell checking was done.

Thank you for your comments.